# SeqLink: A Robust Neural-ODE Architecture for Modelling Partially Observed Time Series

**Futoon M.Abushaqra**                                          *futoon.abu.shaqra@student.rmit.edu.au*
*School of Computing Technologies*
*RMIT University*

**Hao Xue**                                                              *hao.xue1@unsw.edu.au*
*School of Computer Science Engineering*
*University of New South Wales*

**Yongli Ren**                                                          *yongli.ren@rmit.edu.au*
*School of Computing Technologies*
*RMIT University*

**Flora D.Salim**                                                      *flora.salim@unsw.edu.au*
*School of Computer Science Engineering*
*University of New South Wales*

**Reviewed on OpenReview:** *https://openreview.net/forum?id=WCUT6leXKf*

## Abstract

Ordinary Differential Equations (ODEs) based models have become popular as foundation models for solving many time series problems. Combining neural ODEs with traditional RNN models has provided the best representation for irregular time series. However, ODE-based models typically require the trajectory of hidden states to be defined based on either the initial observed value or the most recent observation, raising questions about their effectiveness when dealing with longer sequences and extended time intervals. In this article, we explore the behaviour of the ODE-based models in the context of time series data with varying degrees of sparsity. We introduce SeqLink, an innovative neural architecture designed to enhance the robustness of sequence representation. Unlike traditional approaches that solely rely on the hidden state generated from the last observed value, SeqLink leverages ODE latent representations derived from multiple data samples, enabling it to generate robust data representations regardless of sequence length or data sparsity level. The core concept behind our model is the definition of hidden states for the unobserved values based on the relationships between samples (links between sequences). Through extensive experiments on partially observed synthetic and real-world datasets, we demonstrate that SeqLink improves the modelling of intermittent time series, consistently outperforming state-of-the-art approaches.

## 1 Introduction

Time series analysis of a complex irregular system is regarded as one of the big problems in contemporary data science (Weerakody et al., 2021). Irregular time series occur in many fields, particularly for medical systems where the data is only captured intermittently, leading to gaps in the sequences (Singh et al., 2019). These gaps may sometimes extend over long periods - for example, if a patient misses his/her appointments or fails to use the medical devices at home regularly. However, irregular time series capture dynamic observations without a constant time basis, which makes it hard to model them (Scargle, 1982). Recently, the advances in neural ordinary differential equations (neural ODEs) (Chen et al., 2018), which can produce networks with

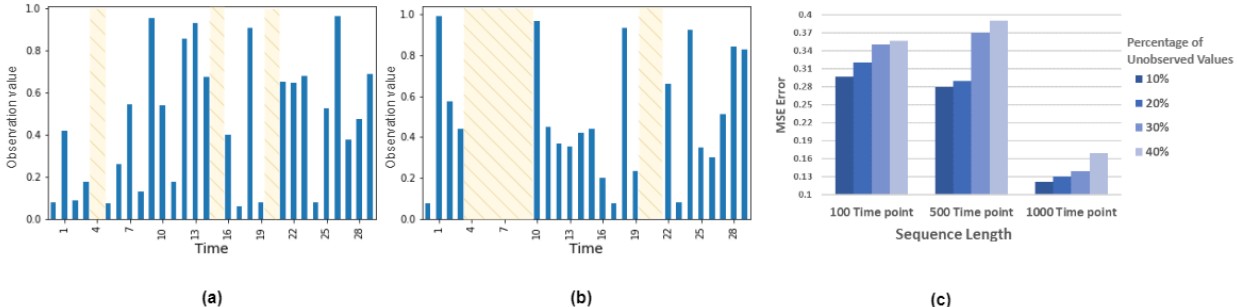

Figure 1: (a) Bumpy irregular time series (with unobserved data highlighted using a yellow-hatched). (b) Intermittent, irregular time series (with unobserved data highlighted using a yellow-hatched). (c) Performance of ODE-RNN model on synthetic intermittent trajectory data with different lengths (100, 500, and 1000 time points) and varying levels of sparseness. The results show that ODE-RNN model is influenced by both the length of the sequence and the sparsity level (time lapse between observations).

continuous hidden states, have become fundamental models for handling irregular sequences. ODEs describe the evolution in time of a process that depends on one variable (initial condition) (Chen et al., 2018); hence, for an ODE-based model, the continuous trajectories of the hidden state are described by just one variable (either the initial value or the last observed value). Since this trajectory represents the data, having the best possible representation of the hidden state is important. Thus neural ODEs are effective but may not provide the optimal representation for the whole sequence, especially when using sequences with long time lapses between observations (e.g. data from medical records). In such cases, the hidden states count on the one initial value for a long time.

To illuminate the limitations of ODE-based models, we describe two distinctive irregular time series patterns: (a) bumpy series (Figure 1.a), which are characterised by frequent and short unobserved intervals, and (b) intermittent series (Figure 1.b), known for having long gaps with numerous unobserved values over extended periods, as outlined in (Zhang et al., 2021). Subsequently, we investigate the behaviour of the ODE-based model under the influence of these irregular time series patterns by conducting a set of experiments on synthetic intermittent datasets with varying levels of sparsity (sets of consecutive unobserved values) as described below in Section 5.1. We study the ODE-based model's performance on sequences of different lengths -using ODE-RNN as an exemplar model. Specifically, we assessed the model forecasting accuracy for sequence lengths of 100, 500, and 1000 time points, each with varying sparsity levels ranging from 10% to 40%.

The results, illustrated in Figure 1.c, show that both sequence length and the degree of sparsity significantly impact the prediction accuracy. However, the main challenge lies in having consecutive unobserved values for a long time. The ODE-based model shows a consistent pattern across different sequence lengths, indicating an inverse relation between the percentage of unavailable observations and prediction accuracy. For example, when 30% of the data is unavailable, the error rate increases by an average of 17.8%, 32%, and 14% for sequence lengths of 100, 500, and 1000 time points respectively, compared to the results when only 10% of the values are unavailable. These results indicate that the ODE representation of continuous unobserved data may vary based on the amount of available data in relation to the sequence length. In the case of longer gaps, the model may not effectively capture the underlying dynamics of irregular time series. This behaviour might be related to the fact mentioned before that ODE-based models rely on a single initial state.

To overcome the limitation of ODE-based models on modelling long intermittent sequences, we present a novel ODE-based architecture (SeqLink) that does not rely on one trajectory (generated by the last available observation) to represent the data; it also produces more generalised hidden trajectories using information learned from similar samples. Our architecture comprises three major parts: (1) an ODE auto-encoder to learn a better representation of the data by employing an encoder to construct hidden trajectories based on ODE; (2) an attention module to categorise the learned representations based on the relations between

samples; (3) a new ODE-based model (Link-ODE) designed and used to model time series by integrating all the categorised learned hidden trajectories from various samples and providing a continuous effective representation for the sequences. The contributions of this work are as follows:

- We have demonstrated how traditional ODE-based models may yield inaccurate predictions and less effective hidden representation when applied to data with a high level of sparsity. Additionally, we show how they are affected by the length of the time lapse between observations. These experiments also serve as the motivation for this work.

- We have proposed a novel approach that utilises diverse ODE-based hidden trajectories to provide an unrestricted representation for unobserved data, enabling us to maintain a good continuous representation over a long period.

- Our proposed method, SeqLink, achieves improved performance over other recent models for time series forecasting on both multivariate and univariate datasets. Additionally, our results demonstrate that SeqLink enhances the latent space representation of unobserved time series and improves prediction accuracy.

## 2 Related Work

Irregularity issues (also known as partially-observed time series) are related to non-uniform intervals between observations (Kidger et al., 2020; Weerakody et al., 2021). In a regular time series, the data follows a specific temporal sequence with a regular interval - for example, samples may always be observed daily. By contrast, in an irregular time series, samples are observed at unevenly spaced intervals. This issue is common for data captured from humans (such as medical data and data on human behaviour), where the system depends on people's commitment (Scargle, 1982; Zhang et al., 2021; Lipton et al., 2016). Irregularity may also be caused by capturing data from heterogeneous sources and sensors (Deldari et al., 2020; Abushaqra et al., 2021; Almaghrabi et al., 2022). However, irregular sampling does not fit with standard machine learning models that assume fixed-size features (Narayan Shukla & Marlin, 2021). Up until the last few years, there were a number of traditional techniques used to handle irregularity, including analysing fully observed samples, performing a features analysis rather than a temporal analysis, or re-sampling and imputation (Zhang et al., 2021; Singh et al., 2019); these methods can destroy temporal information and dependencies.

Although recurrent neural network (RNN) (Robinson & Fallside, 1987; Werbos, 1988) shows outstanding performance in modelling temporal data, it does, on the other hand, assume both fixed gaps between observations and fully observed samples. Recently, with the development of neural ordinary differential equations (neural ODEs) (Chen et al., 2018) in 2018, more effective models have been proposed for irregular data. Neural ODEs is a continuous-time model that defines a latent variable $h(i)$ as the solution for an ODE initial value problem. Rather than specifying a discrete sequence of hidden layers, a continuous representation became possible using the parameterisation of the derivative. To utilise this advantage of the hidden state in neural ODEs, recent models like ODE-RNN and latent ODE (Rubanova et al., 2019) have presented a continuous-time latent state, where the formation of the dynamics between observations is not predefined. These models define the state between observations to be the solution to an ODE, while normal RNN hidden cells are used to update the hidden state at each observation. Therefore the trajectories of the hidden state between observations are defined by the last observed value. As a particular case of the ODE-RNN model, (De Brouwer et al., 2019) provided the GRU-ODE-Bayes model, which includes a continuous-time version of the GRU (GRU-ODE) and a Bayesian update network to handle the sporadic observations. The model combines GRU-ODE and GRU-Bayes, where the first one is used to update the hidden state $h(i)$ in continuous time between observations, and the second is responsible for transforming the hidden state based on the new observation. In recent years, many models based on differential equations (DE) have been presented. These models, including work by Kidger et al. (2020); Morrill et al. (2021); Herrera et al. (2021); Jia & Benson (2019) and others, have enriched our comprehension of DE behaviour and demonstrated enhanced performance across various scales.

To tackle the challenge of modifying the hidden trajectories based on newly received data, the controlled differential equation (CDE) and the neural rough differential equations (RDE) were introduced (Kidger et al.,

2020; Morrill et al., 2021). In contrast to ODE, which primarily relies on their initial states with limited provisions for adjustments, CDE updates the driven value of the ODE equation, denoted as $ds$, by utilising a matrix vector represented as $dX_s$. Therefore the solution of CDE depends continuously on the evolution of $x$ (driven by the control $X$). Neural RDE is an extended neural CDE where, in order to increase memory efficiency, especially for long sequences, the data is modelled without embedding the interpolated path. In recent work by Iakovlev et al. (2023), the authors also focused on faster modelling for long sequences; they provided multiple shooting framework for latent ODE models that works by splitting trajectories of neural ODEs into short segments, optimising them in parallel to facilitate efficient training. Another notable contribution is the work by Schirmer et al. (2022), where they introduced continuous recurrent units (CRUs). These units integrate a linear stochastic differential equation (SDE) within an encoder-decoder framework, using the continuous-discrete Kalman filter to ensure smooth transitions between hidden states and an effective gating mechanism. It is worth noting, however, that the focus of these improved models has primarily been on either facilitating faster learning or enhancing the representation of long sequences. In contrast, our focus is directed towards generating a stable representation for irregular data sets characterised by longer gaps.

Researchers have also explored irregular time series modelling by combining ODE-based models with attention mechanisms (Narayan Shukla & Marlin, 2021; Jhin et al., 2022; Yuan et al., 2022) and long short-term memory (LSTM) networks (Lechner & Hasani, 2020). Furthermore, ODEs have recently been applied in various fields, for instance, in (Yan et al., 2020), where the authors studied the robustness of the neural ODEs and proposed the time-invariant steady neural ODE (TisODE). Their model was then applied to an image classification task by removing the time dependence of the dynamics in an ODE (Garsdal et al., 2022). Additionally, efforts have been made to reduce the high computational overhead caused by the ODE-based models. Habiba & Pearlmutter (2020) redesigned RNN architectures such as GRU and LSTM using ODE, resulting in GRU-ODE and LSTM-ODE models that reduce the computation costs. The models leverage ODE solvers to compute hidden states and cell states, thus substantially reducing the computational cost of additional encoding and decoding used in the previous models (such as Latent-ODE and ODE-RNN). In a recent development, Zhou et al. (2023) introduced the LS4 generative model. LS4, short for latent state space sequential sampler, is designed to capture and generate sequences of data by incorporating latent variables that evolve according to a state space ODE. This model overcomes limitations in existing ODE-based generative models, especially for sequences with sudden changes. LS4 demonstrates enhanced performance and faster training. Another recent contribution Chowdhury et al. (2023) introduced a method for self-supervised learning on irregular multivariate time series. This approach employs contrastive learning and data reconstruction tasks, maintaining the native irregularity of the data. Additionally, it incorporates a time-sensitive data reconstruction task, masking a fixed duration of data instead of a fixed number of observations to ensure tractable reconstruction across regions with varying sampling densities.

As numerical integration, which is used to approximate the solutions of ODEs using numerical methods, significantly influences the model performance, it remains a subject of ongoing investigation. Zhu et al. (2022) explored neural ODEs and their interplay with numerical integration, revealing how neural ODEs approximate certain equations during training. Ott et al. (2020) analysed the connection between differential equations and ResNet, highlighting the strong link between ODE-based models and numerical solvers. Krishnapriyan et al. (2022) conducted experiments to demonstrate the impact of numerical solvers on neural ODEs and proposed a convergence test to select suitable solvers for continuous dynamics.

## 3 Preliminaries

In this section, we introduce the fundamental concepts of ODE-based models, along with the notion of ODE solvers. These concepts serve as the foundation for our proposed methodology. The notations used in the paper are summarised in Table 1.

**ODE:** ODE is a mathematical equation that describes the rate of change of a function with respect to an independent variable. In the context of time series data, ODEs capture the dynamics and relationships between variables over time. A simple form of an ODE is given by: $\frac{dx(t)}{dt} = f(x(t), t)$, where $x(t)$ represents the state of a system at time $t$, and $f(x(t), t)$ defines the rate of change of $x(t)$ at a given time point.

Table 1: Symbols and Notations.

| Notation | Description |
| --- | --- |
| $K$ | A size of data of time series sequences. |
| $k$ | An individual time series sequence extracted from the dataset, denoting a sample sequence within the dataset. |
| $D$ | The dimensionality of each time series, referring to the number of features it encompasses. |
| $t$ | A time index indicating the chronological order of measurements within a time series, starting from 1 to $n$. |
| $n$ | The final time point within a time series. |
| $i$ | A specific time point within the time index $t$. |
| $t_i$ | A time point where $t = i$. |
| $\mathbf{x}$ | The actual observed values of a time series. |
| $x_i$ | The actual observed value at time $i$ of a time series. |
| $m$ | A mask matrix indicating the availability of observations. |
| $m_i$ | A mask value for a specific time point, with a value of either 0 or 1, indicating the presence or absence of a measurement at time point $i$. |
| $h_i$ | A hidden state representing the internal activation of a neural network layer at time point $i$. |

**ODE Solvers:** Solving an ODE involves finding the solution $x(t)$ that satisfies the given differential equation. ODE solvers are numerical methods used to approximate the solution of an ODE over a specified time interval. One common approach is the ODESolve method, which numerically integrates the ODE using discrete time steps. Given an initial condition $x(t_0)$, an ODE solver approximates the values of $x(t)$ at subsequent time points.

**Neural ODEs:** Neural ODEs are an extension of traditional ODEs, where the function $f(x(t), t)$ is parameterized by a neural network. Neural ODEs allow us to model complex and continuous-time dynamics in a data-driven manner. The formulation of a neural ODE is given by:

$$\frac{dh(t)}{d_t} = f_\theta(h(t), t), \quad where \ h(t_0) = h_0 \tag{1}$$

$$h_0, \cdots, h_n = ODESolve(f_\theta, h_0, (t_0, \cdots, t_n)), \tag{2}$$

where $f_\theta$ is a neural network function with learnable parameters $\theta$, and $h(t)$ denotes the state of the system at time $t$. Given a function $f_\theta$ and an initial condition $h_0$ the ODESolve function is used to solve the differential equation and compute the values of $h$ at sequence time points $t_0, \cdots, t_n$.

**ODE-RNN:** is a model that combines the strengths of ODEs and RNNs to model irregular time series data. In the ODE-RNN framework, the hidden state $h_i$ between observations is defined using solutions to an ODE. While at observations, the hidden state is updated using an RNN cell, ensuring that both continuous-time dynamics and observation-specific information are captured. The ODE-RNN formulation is given by:

$$h_i = \begin{cases} RNNCell(h_{i-1}, x_i) & \text{if } m_i = 1 \\ ODESolve(f_\theta, h_{i-1}, (t_{i-1}, \cdots, t_i)) & \text{if } m_i = 0, \end{cases} \tag{3}$$

where $h_i$ represents the hidden state at time $t_i$, $x_i$ is the observation at time $t_i$, $m_i$ is a mask value indicating if the observation is available ($m_i = 1$) or not ($m_i = 0$), and ODESolve is an ODE solver that integrates the ODE using the given function $f_\theta$, an initial state $h_{i-1}$, and a sequence of time points $(t_{i-1}, \ldots, t_i)$.

## 4 Methodology

### 4.1 Problem Statement

We consider modelling $K$ sporadically observed time series with $D$ dimensions. For example, data from $K$ samples (e.g. patients) where $D$ variables are potentially measured at a specific time point $t_i$. Each time series is measured at time points $t = (1, 2, .., n)$. The values of the observations are defined by a value matrix $\mathbf{x} \in \mathbb{R}^{n \times D}$ and a mask matrix $m$ of size $(n \times D)$: $m \in (0, 1)$ to indicate if the variable is observed ($m_i = 1$) or not ($m_i = 0$) at each time point $i$. We assume a specific time series to be sporadically sampled when some samples $x$ have $m$ equal to (0) at one or more time points. The goal is to model the sporadic time series effectively by finding the best continuous latent representation $h$ for the entire sequences.

### 4.2 Overview of SeqLink

In this article, we present a novel system for modelling irregular time series data, aiming to derive generalised continuous hidden representations for unobserved values. As shown in Figure 2: our model comprises three key components: **(1) ODE Auto-Encoder:** This component uses neural ODEs to learn optimal hidden representations for each sample. It takes datasets as input, employing neural ODEs to capture continuous hidden trajectories that best represent each sample. Subsequently, it returns the most suitable representation for each sample. **(2) Pyramidal Attention Mechanism:** Designed to delineate correlations between samples, this method maps data with each other. By leveraging the learned representations as input, it discerns, for each sample, the most relevant representations of other samples. It then sorts these representations based on their relationships to each sample. **(3) Link-ODE:** A generalised ODE-based model tailored to modelling partially observed irregular time series. By utilising the best-hidden trajectories to fill in gaps in the data, this model incorporates learned latent states from another related sample alongside sample-specific latent states to represent each sample effectively. The remaining parts in this section will address each module of SeqLink.

### 4.3 ODE auto-encoder

Inspired by the idea of the denoising auto-encoder (AE) (Vincent et al., 2010), which utilises a corrupted input to train the encoder in order to obtain a high-quality embedding, our ODE-based auto-encoder aims to identify the optimal hidden representation (ODE hidden trajectory). The denoising AE is a type of neural network that learns to denoise data by encoding corrupted inputs and then reconstructing the original data. In a similar vein, our ODE-based AE seeks to minimise the reconstructing error between the original data values $x$ and the decoder output y through the identification of the ODE hidden trajectory. However, our ODE AE deviates from the traditional denoising AE objective, as our primary goal is to obtain a robust and effective latent representation of the data rather than focusing on explicitly denoising it. Therefore, the task is reconstructing $x'$ (corrupted $x$) to the original data $x$ and then obtaining the learned data representations that yield the best result. The proposed ODE AE is presented on the left-hand side of Figure 2, and consists of the following steps:

- Generate $x'$ by corrupting observed $x$ in a time series $k$ using a cut_out function based on a specific number of points to be removed from the timeline. The cut_out function removes the data points by setting them to zero in both the value and mask vectors.

- Each corrupted $x'$ is processed by the ODE-RNN encoder to learn the hidden representation $h_i$ at each time point $t = i$. See Equations (4) and (5). Where the RNN function is used to update the hidden state $u_i$ at observation time $i$ for observation $x_i$. While ODE solver is to solve ODE and get state $u_i$ at time $t_i$ when there are no observations (the time between $i - 1$ and $i$, as Equations 4) as described in Section 3. In other words, Equation 4 is used to find the hidden state for the observations, and Equation 5 is used to find the hidden state between the observations (the gaps).

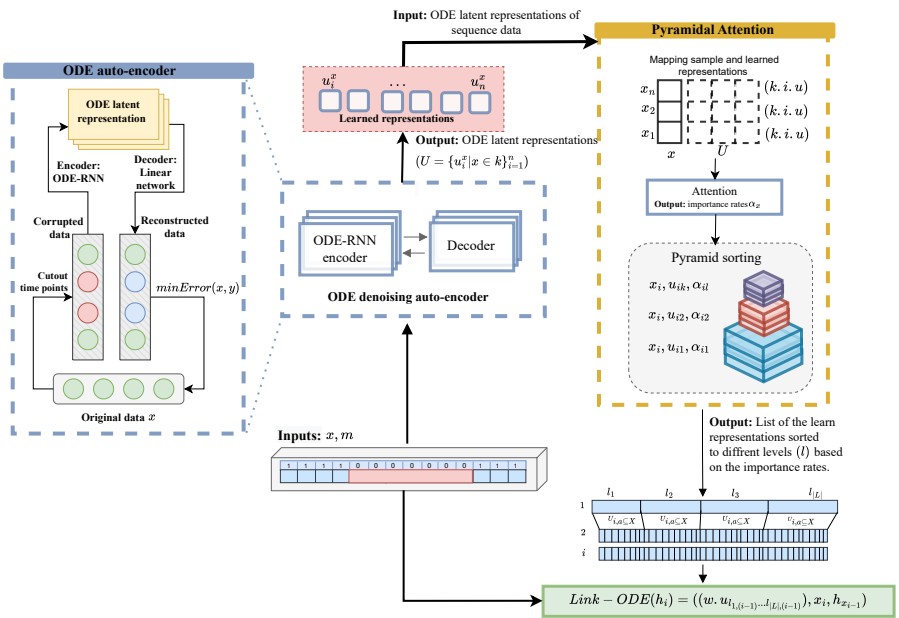

Figure 2: The architecture of SeqLink model with auto-encoder that generates the learned representation for each sequence, pyramidal attention module to sort the representation based on the correlations between samples, and Link-ODE to provide a continuous effective representation for the sequence based on the learned information, where $U$ (generated by ODE auto-encoder) is a set of the best-learned representations $U = \{\{u_i^{(1)}\}_{i=1}^n, \{u_i^{(2)}\}_{i=1}^n, \cdots, \{u_i^{(k)}\}_{i=1}^n\}$ for each sample $k$ at all time points $t_i$, $\alpha$: the importance weights for each latent representation. $l$: a level from 1 to $|L|$ used to sort the learn representations.

$$u_i = ODE\_RNN(x_i) = RNNCell(u_{i-1}, x_i) \tag{4}$$

$$u_i = ODE\_RNN(t_i) = ODESolve(f_\theta, u_{i-1}, (t_{i-1}, \cdots, t_i)) \tag{5}$$

- The latent representations are solved back and decoded to the data space (to undo the effect of the corruption process), where a decoder model of linear sequence layers generates the predicted data $y$.

- The generated data $y$ is compared to the original $x$ with the goal of minimising the re-generating error between $y$ and $x$ as $\arg\min_\rho Loss(X, Y)$, where $\arg\min$ represents the argument (or value) of $\rho$ that minimiSes the function $Loss(X, Y)$, $\rho = w, w', b, b'$ is the set of learning parameters to be optimised for the encoder and decoder, and $Loss(\cdot)$ is the loss function used to measure the similarity between $x$ and $y$. Finally, the learned hidden trajectories of each sample are saved for the next phase as a set of $U = \{\{u_i^{(1)}\}_{i=1}^n, \{u_i^{(2)}\}_{i=1}^n, \cdots, \{u_i^{(k)}\}_{i=1}^n\}$.

## 4.4 Pyramidal Attention

We use the previously learned hidden states $U$ to define a set of latent representations for each sequence based on the correlation between samples. Hence, we find the attention score between the samples and the learned representations from the auto-encoder. As shown in Figure 2, we first map the original data ($x$) to the learned representations ($u$) by embedding both vectors as Equations 6 and 7, where $\varphi$ refers to the embedding layer and $\theta$ represents the learning weights.

$$e_x^{ik} = \varphi_x(x_i^k, \theta_{\varphi_x}) \tag{6}$$

$$e_u^{ik} = \varphi_u(u_i^k, \theta_{\varphi_u}) \tag{7}$$

Next, a concatenate layer combines both vectors as $(S_x)$ as Equation 8, where $\theta$ is a set of learning parameters. This is followed by an attention layer that defines an attention score to find the importance rate $\alpha_x$ between each $x$ and $u$ using Softmax function as the following formula:

$$S_x = (e_x^{ik} \oplus e_u^{ik}) \cdot \theta \tag{8}$$

$$\alpha_x = \frac{exp(S_x)}{\sum_k^K exp(S_x)} \tag{9}$$

The assigned importance weights $\alpha_x$ are used to generate a set of hierarchical levels of related latent representations as $\{l_1, l_2, \ldots, l_L\}$, where each $l_j$ represents a subset of the hidden states from $U$, i.e., $l_j = \{u \subseteq U\}$. At this stage, we use pyramidal sorting for each sample and corresponding learned representations according to the attention importance weights $\alpha_x$ (as shown in Algorithm (1)). The input consists of the importance weights obtained through the attention layer, the learned representation for each sample, and the total number of levels $L$, which determines the pyramid height. The output is a list of sorted representations for each sample, defining the categorisation of entities based on their importance scores.

The pyramidal attention mechanism in our model sorts learned representations based on correlations for each sample, offering advantages over selecting a single best representation. This sorting ensures that the final representation for an unobserved value incorporates information from multiple related latent representations, reducing reliance on a single sample and enhancing the model's generalisability. By constructing a relevance pyramid, higher levels contain representations with higher importance rates, while lower levels contain representations with lower rates, allowing comprehensive consideration of information from all samples, avoiding over-reliance on individual samples and reducing the influence of outliers or noise. Moreover, the pyramidal mechanism introduces a negligible additional computational cost compared to using only attention, as it involves the same standard operations.

---

**Algorithm 1: Pyramidal sorting**

---

**1 Input:** Attention_weights (rates) $\alpha$ , Learned_representations $U$, number_of_level $L$

**2** $sorted\_u_k = []$      `// sorted weights for one sample`

**3** $sorted\_u = []$      `// sorted weights for all sample`

**4 for** $k$ **in** $1, 2, .., K$ **do:**

**5**      $rates = \alpha[k,]$

**6**      **for** $l$ **in** $1, 2, .., L$**:**

**7**          $MeanV = \sum rates/K$

**8**          $Current\_L = [rates <= MeanV]$

**9**          $Current\_U = U[Current\_L].mean()$

**10**          $rates = [rates > MeanV]$

**11**          $sorted\_u_k.append(Current\_U)$

**12 end for**

**13** $sorted\_u.append(sorted\_u_k)$

**14 Output:** $sorted\_u$

---

### 4.5 Link-ODE

Using the learned hidden representations, we define a Link-ODE network (shown in Figure 3) that in generating a continuous representation for the unobserved part not only considers the last observed value but also uses the ODE trajectories learned from different samples. The implementation of the Link-ODE is represented in Algorithm 2. An ODE solver is used to define the hidden state at time $i + 1$ based on the last observed value (initial value), while another ODE solver is used to define a hidden state trajectory for $i + 1$ for other samples (ODE solver from the ODE AE discussed in section 4.3). Finally, an RNN cell is used to find the hidden state when there is an observation based on hidden states from the ODE solver for the current sample, the hidden states of other samples, and the current value of $x_i$. In other words, when $m_i = 1$ the hidden state is defined based on the last hidden state and the current value of $x_i$ for each $x_i$, and

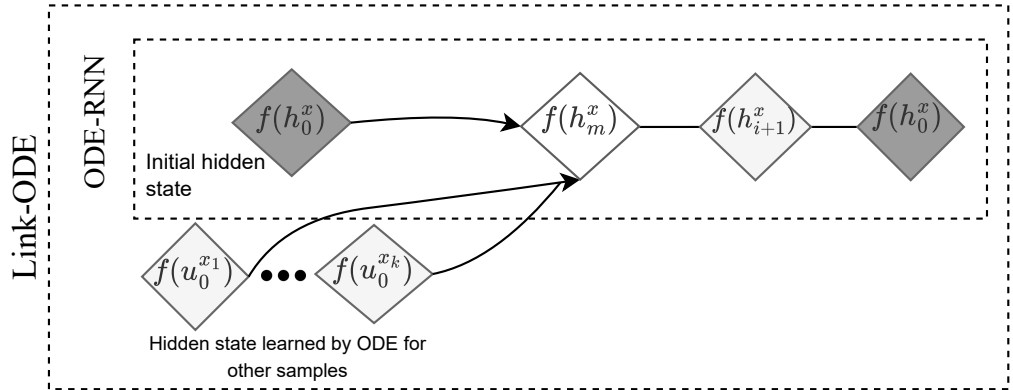

Figure 3: Architecture of Link-ODE model (the green part in Figure 2). Continuous-time modelling for irregular samples, where $h_0$ is the initial state of a time point $t_i$, $f(\cdot)$ is an ODE cell (update function) to solve the ODE for $h$, and $u_0^x$ is the previously learned trajectories from other samples $x_k$ at the same time point.

when $m_i = 0$ the hidden state is defined based on the ODE solver for the last hidden state and the hidden states learned from other samples. The previously learned states are combined and multiplied by a specific weight based on their level of correlation to the sample ($p$), as expressed in the following Equations (10,11):

$$p = w \cdot l_1, w \cdot l_2, \cdots, w \cdot l_L \tag{10}$$

$$h_{i+1} = RNNCell(h_{i+1}, p, x_{i+1}) \tag{11}$$

$p$ is a matrix encompassing all levels, where each $l$ corresponds to latent representations assigned to that specific level. The weight values, denoted by $w$, are established according to the levels, wherein latent representations at a given level $l$ receive higher weights when positioned at the pyramid's apex. This apex placement signifies a heightened relevance to the current value of $x$, as determined by the attention layer. After obtaining a continuous representation that encapsulates the learned relationships and dynamics within the data, a conventional output dense layer is employed, where the representations are then fed to generate the final output.

---

**Algorithm 2: Link-ODE**

---

**1 Input:** Data points $x$

**2 Output:** $h_N$

**3 for** $i$ **in** $1, 2, .., n$ **do:**

**4**    $\ddot{h}_{i+1} = ODESolve(f_\theta, h_{x_i}, (t_i, t_{i+1}))$ // Solve ODE to get state at $t_{i+1}$ based on last hidden state of $x_i$.

**5**    $\bar{h}_{i+1} = ODESolve(f_\theta, h_{x_i^k}, (t_i, t_{i+1}))$ // Solve ODE of $h_{i+1}$ of other sample in $K$.

**6**    $h_{i+1} = RNNCell(\ddot{h}_{i+1}, \bar{h}_{i+1}, x_{i+1})$

**7 end for Pass;**

---

## 5 Experiments

### 5.1 Datasets

**Synthetic data (Gaussian trajectories):** To evaluate the performance of our proposed model and analyse the model's robustness, we generated three synthetic datasets with 1000 samples of periodic trajectories and varying sequence lengths using the standard Gaussian function. The standard Gaussian function, denoted as $N(0, 1)$, is a specific form of the Gaussian distribution with a mean (µ) of 0 and a standard deviation (s) of 1. It is a fundamental probability distribution to model random variables with continuous outcomes.

Using that function, we generate the following three periodic datasets: (1) a dataset with 100 time points (refer to it as *PeriodicDataset_100*), (2) a dataset with 500 time points (as *PeriodicDataset_500*) and (3) a dataset with 1000 time points (as *PeriodicDataset_1000*). For each dataset, we use a cut-out function to simulate unobserved values by sub-sampling the data and generating sporadic sequences. We generate four sets for each dataset with different percentages of sparsity (10%, 20%, 30%, and 40%). The cut-out function removes a value by setting it to zero in the value tensor ($x$) and mask tensor ($m$). In total, twelve synthetic trajectories are used. This dataset was able to show the effect of sequence length and number of unobserved values on the performance of the models.

**Real-world datasets:** Additionally, we used four benchmark real-world datasets, including:

- Electricity Consumption Load (ECL) [1]: A public dataset that describes the electricity consumption (Kwh) of 321 clients. From the ECL dataset, we used observations for one year and ten sites. We generated sequences with 30 days lag and prediction one day ahead. Since the ECL dataset is a regular time series, we performed a cut-out function on the samples. We randomly cut out 30% of the time points to generate a sporadic series. In total, 3350 univariate sequences were used.

- Electricity Transformer Temperature (ETT) (Zhou et al., 2021)[2]: A multivariate time series that records two years of hourly data. The data has six features used to predict the electrical transformers' oil temperature based on load capacity. We used the available small dataset from 2016 and generated 2000 sequences with lag length of 24 hours and prediction one hour ahead. For the ETT series we also cut out 30% of the observations to generate a sporadic series.

- Weather Data[3]: This data describes local climatological conditions for various US sites. It includes 11 climate features and the target value (a wet-bulb temperature). We built sequences of 7 day lag using hourly data with prediction one hour ahead. We used 2000 sequences and performed a 30% cut-out to generate irregular samples.

- MIMIC-II (PhysioNet Challenge 2012 data) (Silva et al., 2012)[4]: An irregular medical data, describing measurements for patients in ICU. It includes 48 measurements, 37 features, and a binary target value for in-hospital mortality. For our experiments, we used data for 1000 patients. Following (Rubanova et al., 2019), we rounded up the time stamps to one minute to speed up the training process.

## 5.2 Experiment Details

**Baselines:** As our architecture is built based on ODE-RNN (Rubanova et al., 2019), we selected ODE-RNN as our major baseline. We also compared our models' performance to latent ODE (as this is a popular baseline model for irregularly sampled time series)(Rubanova et al., 2019), two classical time series models (RNN and RNN-VAE), and a novel neural DE model (CDE) (Kidger et al., 2020). Along with a recent state-of-the-art forecasting model, TSMixer (Chen et al., 2023).

**Setup:** For all the experiments, we applied a shuffled splitting to divide the data into a training set and a testing set. 80% of the samples were used for training, while the testing set held the remaining 20%. We re-scaled (normalised) the features between (0,1) for each dataset. To generate sparse data for the ECL, ETT, and weather datasets, we randomly cut out samples by setting the value to zero for the value and mask vectors. For all these real-world forecasting datasets, we cut out 30% of the time points in each sample. MIMIC-II is already a sparse dataset, so no cut-out function was performed. For model hyperparameters, to make the experiments fair and consistent, we followed (Rubanova et al., 2019) and chose the hyperparameters that yield the best performance for the original ODE-RNN. We ran both baselines and SeqLink with the exact same size of the hidden state and the same number of layers and units. For the auto-encoder in SeqLink we used ODE-RNN for the encoder and a shallow sequential network of one linear layer to decode

---

[1] https://archive.ics.uci.edu/ml/datasets/ElectricityLoadDiagrams20112014
[2] https://github.com/zhouhaoyi/ETDataset
[3] https://www.ncei.noaa.gov/ data/local-climatological-data
[4] https://pubmed.ncbi.nlm.nih.gov/24678516/

Table 2: Comparison of MSE values (for forecasting) and AUC values (for classification) of SeqLink against various baseline models.

| | Forecasting task (MSE) | | | | | | |
|---|---|---|---|---|---|---|---|
| Dataset | RNN | RNN VAE | Latent ODE | CDE | ODE-RNN | TSMixer | SeqLink |
| Gaussian data | 0.767 | 7.695 | 8.28 | 8.686 | 0.126 | 1.606 | **0.114** |
| ECL | 0.520 | 3.400 | 0.620 | 2.136 | 0.500 | 1.2848 | **0.480** |
| ETT | 0.212 | 0.374 | 0.202 | 0.424 | 0.230 | 0.556 | **0.199** |
| Weather | 0.735 | 11.724 | 0.783 | 0.897 | 0.660 | 1.389 | **0.630** |
| | Classification task (AUC) | | | | | | |
| MIMIC-II | 0.667 | 0.518 | 0.701 | NA | 0.692 | NA | **0.720** |

Table 3: STD for test results over three runs for all baseline models and SeqLink on real-world datasets.

| Dataset | RNN | RNN VAE | Latent ODE | CDE | ODE-RNN | TSMixer | SeqLink |
|---|---|---|---|---|---|---|---|
| Gaussian data | 0.044 | 0.006 | 0.009 | 0.0856 | 6.59E-04 | 0.5061 | 2.24E-04 |
| ECL | 0.0018 | 0.0042 | 0.0024 | 0.009 | 0.0001 | 1.44 | 0.0001 |
| ETT | 0.1833 | 0.0057 | 0.0004 | 0.016 | 0.0081 | 0.1740 | 0.0509 |
| Weather | 0.0208 | 0.2216 | 0.0389 | 0.114 | 0.0274 | 0.1303 | 0.0065 |

the data to the data space and solve back the ODE. While for Link-ODE we used the ODE function of one hidden layer and 100 units. The latent dimension used was 10 for all data sets. The fifth-order "dopri5" solver from the "torchdiffeq" python package was used as the ODE solver. This is the same one used in the traditional ODE-RNN model. We run 200 epochs on a batch size of 200. The same settings are used for all models (SeqLink and the baselines). Finally, We used mean squared error (MSE) to evaluate the prediction performance and area under curve (AUC) for the classification task.

**Resource and Training:** We used Adam optimizer and a learning rate of 0.01. The experiments were run on a desktop with an NVIDIA GeForce MX230.

**Implementation:** We build our code on the publicly available code for ODE-RNN at (`https://github.com/YuliaRubanova/latent_ode`), using PyTorch. For the baselines (RNN-VAE, Latent ODE and ODE-RNN) we follow the implementation available at `https://github.com/YuliaRubanova/latent_ode`). While for the CDE model, we follow the implementation available at (`https://github.com/patrick-kidger/NeuralCDE`). For TSMixer we follow the implementation available at (`https://github.com/ditschuk/pytorch-tsmixer`). Our code is available at (`https://github.com/FtoonAbushaqra/SeqLink.git`)

## 5.3 Model Performance

Tables 2 presents the results of the SeqLink model against different baselines. The first part shows the MSE values for forecasting tasks on real-world data and the average performance for synthetic data (Gaussian trajectories), while the last row sets out the AUC results for the classification task performed on the MIMIC-II dataset. The results are the average of three runs with different random seeds used to initialise the model parameters. The best performance against each dataset is highlighted in bold. In general, our method outperforms the baselines for all real-world datasets. Among the baseline models, the ODE-RNN model achieved the best performance for the forecasting task. By contrast, RNN-VAE and latent ODE failed to model these partially observed datasets. Although the CDE model was established to increase the representation learning capability of the neural ODEs, it required a longer processing time and failed to solve forecasting tasks. In the classification task, SeqLink has the highest accuracy on the MIMIC-II data, while the Latent ODE model had the best performance among the baseline models. For completeness, since we report the average performance across three runs, we present the standard deviation error for each dataset in Table 3.

Additionally, we present a comprehensive analysis of the performance of SeqLink and the baseline models using the synthetic trajectories to highlight the effectiveness and the behaviour of the model. The detailed

Table 4: Comparison of MSE values of SeqLink against various baseline models on Gaussian trajectories datasets with different sparsity level.

| Dataset
% of sparseness | | RNN | RNN VAE | Latent ODE | CDE | ODE-RNN | TSMixer | SeqLink |
|---|---|---|---|---|---|---|---|---|
| PeriodicDataset_100 | 10% | 1.127 | 7.535 | 8.390 | 7.106 | 0.297 | 2.952 | **0.280** |
| | 20% | 1.231 | 7.783 | 8.356 | 8.207 | 0.319 | 1.158 | **0.297** |
| | 30% | 1.097 | 7.345 | 8.425 | 8.772 | 0.349 | 1.084 | **0.318** |
| | 40% | 5.099 | 7.21 | 8.404 | 9.215 | 0.356 | 1.645 | **0.307** |
| PeriodicDataset_500 | 10% | 0.106 | 7.855 | 8.334 | 7.312 | 0.028 | 0.945 | **0.027** |
| | 20% | 0.131 | 7.884 | 8.391 | 8.418 | 0.029 | 1.715 | **0.026** |
| | 30% | 0.095 | 7.671 | 8.166 | 9.397 | 0.037 | 1.253 | **0.030** |
| | 40% | 0.128 | 7.565 | 8.072 | 9.948 | 0.039 | 1.646 | **0.032** |
| PeriodicDataset_1000 | 10% | 0.048 | 7.992 | 8.230 | 7.761 | **0.012** | 2.0512 | **0.012** |
| | 20% | 0.054 | 7.901 | 8.209 | 8.187 | 0.013 | 1.669 | **0.011** |
| | 30% | 0.040 | 7.872 | 8.289 | 9.897 | 0.014 | 1.3875 | **0.012** |
| | 40% | 0.052 | 7.731 | 8.094 | 10.02 | 0.017 | 1.765 | **0.015** |

Table 5: STD for test results over three runs on the synthetic datasets.

| Dataset
% of sparseness | | RNN | RNN VAE | Latent ODE | CDE | ODE-RNN | TSMixer | SeqLink |
|---|---|---|---|---|---|---|---|---|
| PeriodicDataset_100 | 10% | 0.0205 | 0.0022 | 0.0178 | 0.1087 | 0.0003 | 1.7898 | 0.001 |
| | 20% | 0.0212 | 0.0586 | 0.012 | 0.0643 | 0.0011 | 0.2244 | 0.0006 |
| | 30% | 0.0196 | 0.0021 | 0.0095 | 0.0399 | 0.0022 | 0.0603 | 0.0002 |
| | 40% | 0.468 | 0.002 | 0.0115 | 0.0355 | 0.0002 | 0.9501 | 2.8E-05 |
| PeriodicDataset_500 | 10% | 0.001 | 0.0007 | 0.0051 | 0.028 | 0.0012 | 0.0236 | 0.0003 |
| | 20% | 0.0017 | 0.0005 | 0.018 | 0.034 | 0.0009 | 0.6892 | 0.0002 |
| | 30% | 0.0016 | 0.0004 | 0.0065 | 0.046 | 0.0007 | 0.1594 | 0.0001 |
| | 40% | 0.0035 | 0.0008 | 0.0003 | 0.066 | 0.0009 | 0.7661 | 0.0001 |
| PeriodicDataset_1000 | 10% | 0.0001 | 7.0E-06 | 0.0058 | 0.1205 | 0.0003 | 0.5511 | 7.0E-06 |
| | 20% | 7.0E-06 | 0.0007 | 0.0004 | 0.057 | 7.0E-07 | 0.2017 | 8.7E-06 |
| | 30% | 5.6E-06 | 0.0001 | 0.0225 | 0.403 | 0.0001 | 0.1155 | 4.6E-05 |
| | 40% | 7.0E-06 | 0.0008 | 0.0004 | 0.025 | 2.8E-06 | 0.5416 | 0.0001 |

MSE values for all models using the synthetic dataset described before are given in Table 4, and the standard deviation errors for three-time runs are presented in Table 5. Remarkably, across all sparsity levels, SeqLink consistently outperforms the baseline. Again, baseline models, including ODE-based models (latent-ODE, CDE), have failed to model this data, especially since the unobserved values are consecutive. However, when examining the performance on synthetic datasets with varying sequence lengths, both ODE-RNN and SeqLink exhibit improved performance for longer sequences. For instance, the average MSE value for *PeriodicDataset_100* using the ODE-RNN model is 0.33, but this figure significantly improves by over 89% to 0.033 for the *PeriodicDataset_500* dataset and further to 0.014 for the *PeriodicDataset_1000* dataset. A similar trend is observed for SeqLink with a more than 90% reduction in MSE values for the *PeriodicDataset_1000* and *PeriodicDataset_500* datasets compared to the *PeriodicDataset_100* dataset. This observed improvement aligns with the intuitive notion that providing a longer sequence for learning equips the model with more information, enhancing its effectiveness compared to shorter sequences with less information.

Furthermore, when considering the impact of sparsity levels, a distinct advantage of SeqLink becomes evident, particularly in datasets with higher sparsity levels (fewer available observations). This advantage is highly explicit when the sequence length is short. In comparison, when examining the performance of SeqLink to the baseline (ODE-RNN), SeqLink consistently outperforms in scenarios with higher sparsity levels, especially when the sequence is short. This is because having long gaps in a short sequence means there is less information (actual observation) available for learning.

Table 6: Percentage improvement when using SeqLink over traditional ODE-RNN. Note the increased benefits as the time lapses (gaps) grow longer, especially when the sequence length is short.

| Percentage of sparseness | 10% | 20% | 30% | 40% |
|---|---|---|---|---|
| PeriodicDataset_100 | 5.5% | 6% | 7% | 14% |
| PeriodicDataset_500 | 1% | 11% | 18% | 17% |
| PeriodicDataset_1000 | 3% | 10% | 7% | 9% |
| Improvement rate | 1% | | | 20% |

Table 6 provides insights into the percentage improvement achieved by utilising SeqLink over the traditional ODE-RNN. For the *PeriodicDataset_100* with 10% sparsity, the performance of SeqLink exhibited a 5.5% improvement over the ODE-RNN. In contrast, this improvement increased to 14% when 40% of the values from the same dataset were unobserved. This suggests that our methodology, which defines hidden states between observations by considering data from different sequences, contributes to a better representation, especially when the available observations are limited. Moreover, both methods performed better for longer sequence lengths. Thus, the evaluation indicates that the model's performance is not solely dependent on the ratio of available to unavailable observations but also, to some extent, on the number of observations available. Additionally, this evaluation further emphasises that hidden states generated using traditional ODE-based models do not provide an effective representation of the whole sequences.

From our detailed analysis of the results for datasets with different lengths, we can draw attention to the following conclusions: (1) The process of modelling irregular sequences is quite challenging for shorter sequences. (2) A model's efficiency on irregular series does not only rely on the percentage of unavailable observations, as the total number of available observations also impacts performance.

## 5.4 Ablation Study

To explore the effectiveness of SeqLink we conducted a series of experiments and tested the following various configurations to define the latent trajectories:

- Unified hidden trajectories: Directly uses all learned hidden representations from the autoencoder without considering the relation between the samples (by removing the pyramid attention module).

- Most related trajectories: Uses only those representations with the highest importance rate from among all the learned representations.

- Least related trajectories: Uses the representations with the lowest importance rates.

The results of the ablation study are presented in Table 7. The performance of the original SeqLink model is superior to the other three configurations. On the other hand, using the most important latent trajectory and the unified representation (underlined) outperforms the least important learned representations that show weak performance, which indicates that the attention layer does learn the relations between samples. Injecting hidden states of unrelated samples lowers the performance of the model, where unrelated samples affect the final representation. On the other hand, relying on all sequences may also introduce unrelated information to the representation but with less effect compared with the previous case. Conversely, relying solely on the best related trajectories provides a very restrictive and biased hidden trajectory. Therefore, we can observe that the original framework of SeqLink outperforms.

## 6 Model Interpretation and Discussion

The total number of levels for the pyramid, denoted as $L$ (as illustrated in Figure 2 and discussed in Section 4.4), is a critical parameter in the SeqLink model. The value of $L$ determines the height of the pyramid, thereby influencing the importance assigned to each correlated sample. To research this parameter, we examined several pyramids with different sizes: (1) shorter pyramids with compacted levels, indicating fewer

Table 7: Average MSE results from ablation studies of three different configurations.

| Dataset | Unified hidden trajectories | Most related trajectories | Least related trajectories | SeqLink |
|---------|------------------------------|----------------------------|-----------------------------|---------|
| PeriodicDataset_100 | 0.339 | 0.320 | 0.338 | **0.318** |
| PeriodicDataset_500 | 0.031 | 0.033 | 0.034 | **0.030** |
| PeriodicDataset_1000 | 0.013 | 0.013 | 0.023 | **0.012** |
| ECL | 0.491 | 0.501 | 0.505 | **0.480** |
| ETT | 0.203 | 0.204 | 0.211 | **0.199** |
| Weather | 0.708 | 0.684 | 0.711 | **0.630** |
| MIMIC-II (AUC) | 0.706 | 0.691 | 0.69 | **0.720** |

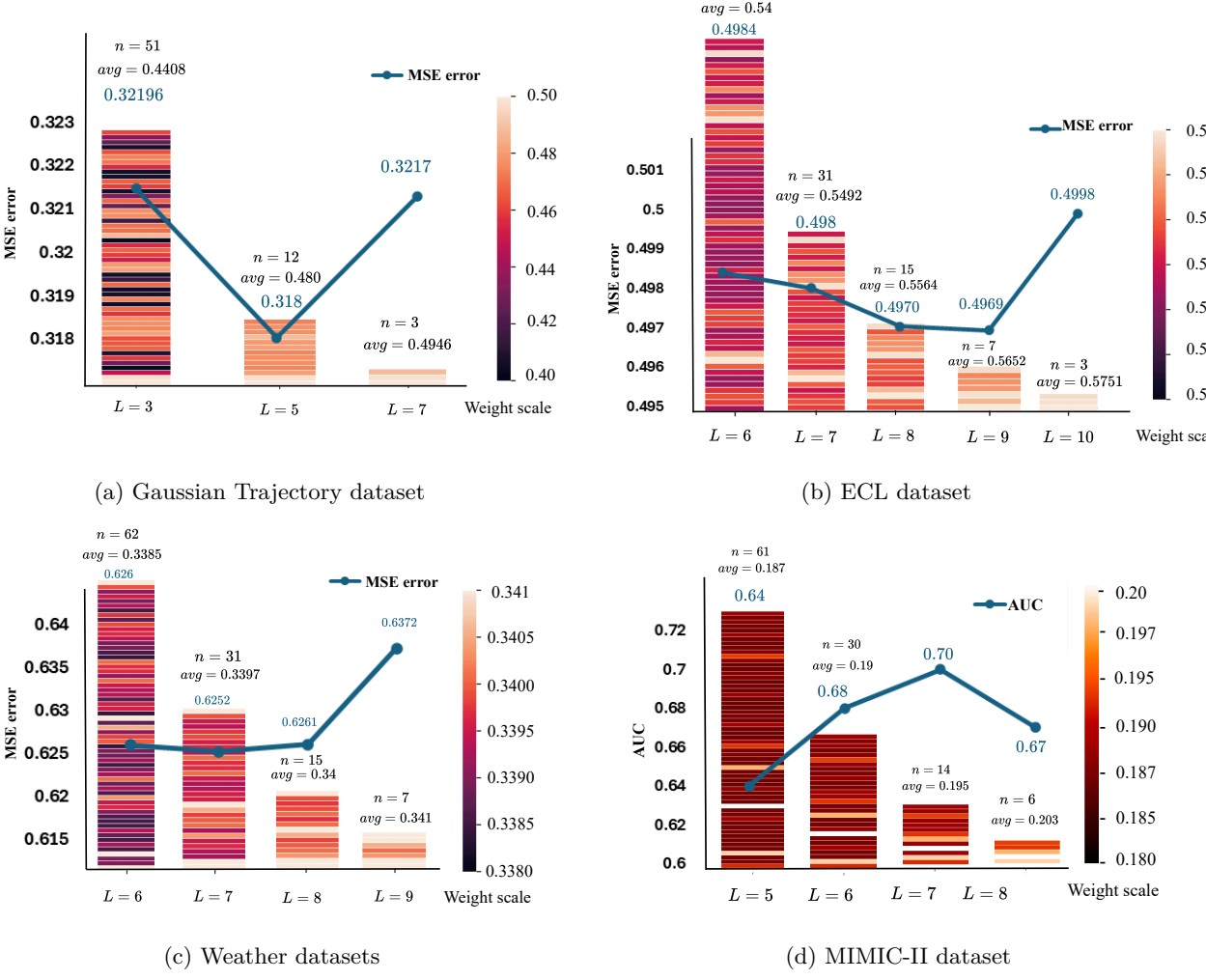

(a) Gaussian Trajectory dataset

(b) ECL dataset

(c) Weather datasets

(d) MIMIC-II dataset

Figure 4: Illustration of the model's behaviour in terms of the attention weights for samples from different datasets using several values of L. Each plot shows the number of samples ($n$) assigned to the topmost layer of the pyramid after using different values of $L$. For each value, we display the average importance rates for these samples ($avg$) and the MSE (or AUC) values achieved by the model (blue line plot). Using a very high or very low value of $L$ affects the model's performance, with the optimal performance consistently achieved at a suitable value of $L$, which also depends on the size and correlation between the samples.

levels with more samples in each, and (2) taller pyramids with loose levels, signifying more levels but fewer samples in each.

Figure 4 presents the importance rates for samples from various datasets, showcasing the top layer of the pyramid at different values of $L$, along with the number of samples $n$ assigned to that layer, the average weight (importance) rate for these samples and the MSE error achieved by the model (blue line). Our findings reveal that highly compacted layers tend to allocate more weight scores to less important samples due to the longer radius between the samples and correlated samples. Consequently, less correlated samples are considered more important. On the other hand, too loose levels focus on very few correlated samples, introducing bias toward a limited set of samples. For instance, when $L = 3$, 51 samples were considered highly related for the Gaussian dataset with an average importance rate of 0.44, while only three were deemed related when $L = 7$, with an average importance rate of 0.49. To see the effect of that on the model performance, we can notice that the MSE values (presented as a blue line plot) improve when having a larger value of $L$ but then decrease for a very large value of $L$. In the case of the Gaussian dataset, the best performance achieved was 0.31 for $L = 5$, while MSE values were 0.321 when using $L = 3$ or $L = 7$. Similar trends could be noticed for other datasets where the performance improved for a suitable value of $L$. For MIMIC-II, we report the AUC, where the model also showed improvement for $L = 7$ with an accuracy of 70%, while it dropped to 67%, 68%, and 64% for $L = 8, 6$, and 5 respectively. Notice that the value of $L$ is also determined by the total size of the dataset, where we experimentally defined the range of $L$ values to explore for each dataset.

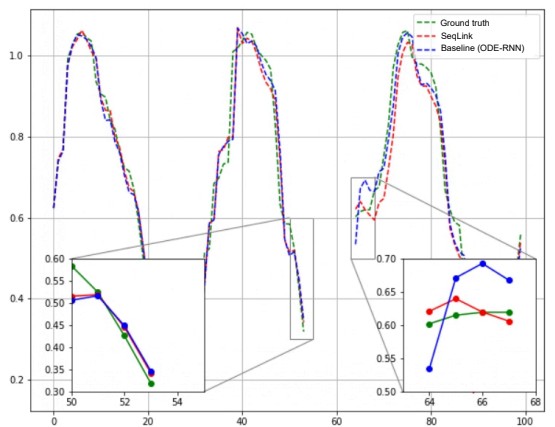
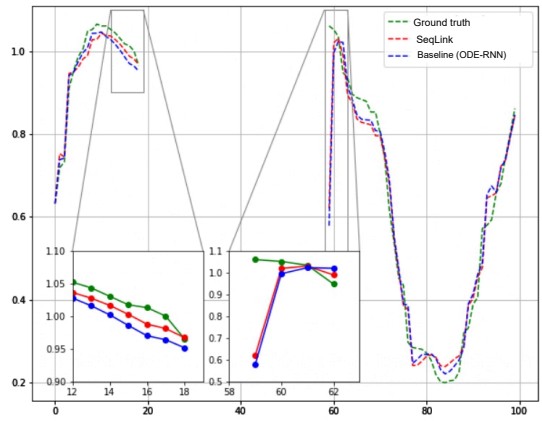

(a) Sequences of a sample with 10% unobserved values     (b) Sequences of a sample with 40% unobserved values

Figure 5: Case Study: Sequences of two samples from *PeriodicDataset_100* dataset with 10% unobserved values (a) and 40% unobserved values (b). Actual values are shown with the green dashed line, SeqLink predictions are shown with the red dashed line, and baseline (ODE-RNN) predictions are shown with the blue dashed line. SeqLink outperforms the baseline and can give more accurate predictions for the values that appear after the gap.

Furthermore, to demonstrate the robustness of the hidden representation of SeqLink, two case studies are investigated in Figures 5.a and b. The figures show two samples from *PeriodicDataset_100* dataset, one with 10% unobserved values, and the other with 40% of its values unobserved. Each figure illustrates the sequence of the actual values, the SeqLink predictions, and the ODE-RNN predictions. We have highlighted the predictions at different time points for more clarity. The generated sequences show that our model not only outperforms the overall sequence prediction but also gives a closer prediction for the values that show up directly after a gap, even when the gaps are longer. For instance, in Figure 5.b, the difference between the predicted and actual values for the time points 58 to 63 is about 8% less when using SeqLink as compared with ODE-RNN. More case studies are presented in Figure 6. Although the baseline is better at some time point (Figure 6.d), our proposed method outperforms most sequences. These cases implicitly prove that

our model's hidden representations for the unobserved time points are more related than the hidden state calculated in the traditional ODE-RNN.

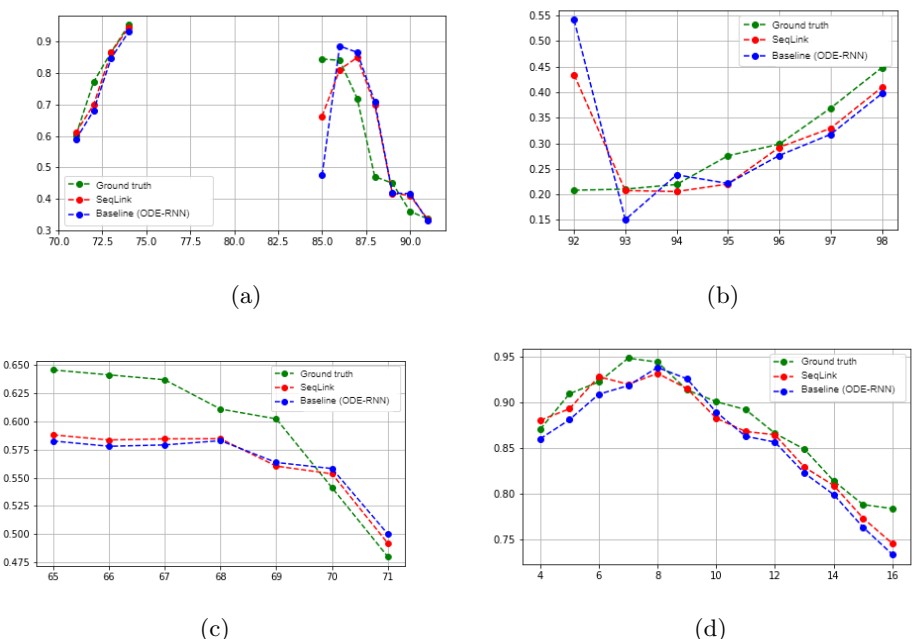

Figure 6: Extra examples of the predictions generated by ODE-RNN and SeqLink.

**Test of significance:** Finally, we employed the Wilcoxon rank-sum test to determine whether the proposed model significantly outperformed the baseline model. The results from the previous examples demonstrated a statistically significant difference in performance (p=0.0226) on average, indicating that the proposed model produced superior results.

## 7 Conclusion

In this work, we have proposed a novel ODE-based model to generalise hidden trajectories and analyse irregular data. We first explored the behaviour of the ODE-based model on partially observed sequences with different lengths and time lapses. In response to the issues found, we presented the SeqLink model that can build unrestricted ODE representations for the unobserved values and maintain good continuous representations over a long time. The results of extensive experiments on different tasks and datasets show that SeqLink outperforms other state-of-the-art models. Our ablation study and case studies show that the framework of our model is able to improve the representation of the unobserved values for partially-observed time series. Nevertheless, our proposed model was built on the ODE-RNN model as a basic ODE-based model. For future work, we plan to explore and investigate other ODE-based models using our proposed framework.

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
