# OpenReview forum: "SeqLink: A Robust Neural-ODE Architecture for Modelling Partially Observed Time Series"
_TMLR — Accepted by TMLR_

### Review · Reviewer_kkXW · 2024-04-28

**Summary Of Contributions:**

The work proposes SeqLink a neural-ODE based architecture for modelling multiple irregular time-series data. The proposed approach consists of a denoising auto-encoder based on ODE-RNNs, a pyramidal attention module which sorts and correlates the learned representation across different samples, and a link ODE that takes learned representation from multiple samples as inputs. The work compares the proposed approach against other baseline models for irregular time series including RNN VAE, Latent ODE, CDE, ODE-RNN and SeqLink shows competitive performance against the baseline models.

**Audience:**

Yes

**Broader Impact Concerns:**

N.A.

**Claims And Evidence:**

No

**Requested Changes:**

Please make changes to address the weakness of the work, especially the weakness on presentation.

**Strengths And Weaknesses:**

Stregnths
1. The work studies an interesting problem of modelling multiple time series samples together.
2. Experiment results show the competitive performance against baseline models.
3. The experiments settings are comprehensive with both synthetic and real-world data as well as ablation studies.

Weakness:

Despite the value of the problem the work focuses on and its proposed approach, the presentation of the work needs significant improvement before the reviewer can assess its contributions. The presentation could be improved in the following ways:
1. Clearly define each notation. For example, in Sec. 4.3, $\mu$ is not clearly defined; it is not clear how $S_{x_{i}}$ from Eq.9 is obtained from Eq. 8; in Algorithm 1, the inputs does not appear in the step 5-13, it is not clear where the variable $score$ is from and what is the output of the algorithm.
2. For each component of the model, especially the pyramidal attention, clearly state the input and output.
3. Consolidate some quantitative evaluation tables. For example, Table 2 and Table 3 can be combined into one table.
4. Some most recent works might be missing in the baselines compared against. This work claims superior performance against baseline models on long sequences. Recently, state space models like S4[1] and Mamba[2] shows excellent performance on modelling long sequences and the model should consider comparing against these models. The denoising auto encoder of the work is also similar to some recent works on self-supervised pre-training for time series data [3].

References

[1] Gu, Albert, Karan Goel, and Christopher Ré. "Efficiently modeling long sequences with structured state spaces." arXiv preprint arXiv:2111.00396 (2021).

[2] Gu, Albert, and Tri Dao. "Mamba: Linear-time sequence modeling with selective state spaces." arXiv preprint arXiv:2312.00752 (2023).

[3] Chowdhury, Ranak Roy, et al. "Primenet: Pre-training for irregular multivariate time series." Proceedings of the AAAI Conference on Artificial Intelligence. Vol. 37. No. 6. 2023.

---

> ### Author Response · Authors · 2024-05-07
>
> Thank you for your feedback. Here are the improvements made addressing all your concerns and questions
>
> 1- Equation Notation: The notation has been revised for clarity,  $S_x$ is generated by a concatenation layer between samples and latent representations. Additionally, Algorithm 1 has been enhanced, as detailed on page 8.
>
> 2-  Modules inputs and outputs: The details of each module's inputs and output are clearly mentioned in several parts. A new subsection (4.2) titled "Model Overview" has been added to provide a clearer understanding of the modules' inputs and outputs. Furthermore, Figure 1 has been improved.
>
> 3- Tables Consolidation: Although attempts were made to combine Tables 2 and 3 (4 and 5), the resulting text became too small to be legible for page fitting purposes. Therefore, the values are presented in separate tables. However, there is a version of the combined tables where this issue is not present.
> | Dataset         | RNN            | RNN VAE       | Latent ODE    | CDE            | ODE-RNN        | TSMixer        | Model            |
> |-----------------|----------------|---------------|---------------|----------------|----------------|----------------|------------------|
> | Gaussian data   | 0.767 (0.044)  | 7.695 (0.006) | 8.28 (0.009)  | 8.686 (0.0856) | 0.126 (6E-04)  | 1.606 (0.5061) | **0.114 (2E-04)**|
> | ECL             | 0.520 (0.0018) | 3.400 (0.0042)| 0.620 (0.0024)| 2.136 (0.009)  | 0.500 (0.0001) | 1.2848 (1.44)  | **0.480 (0.0001)**|
> | ETT             | 0.212 (0.1833) | 0.374 (0.0057)| 0.202 (0.0004)| 0.424 (0.016)  | 0.230 (0.0081) | 0.556 (0.1740) | **0.199 (0.0509)**|
> | Weather         | 0.735 (0.0208) | 11.724 (0.2216)| 0.783 (0.0389)| 0.897 (0.114)  | 0.660 (0.0274) | 1.389 (0.1303) | **0.630 (0.0065)**|
>
>
> 4- Adding Baselines: Our focus is primarily on irregular datasets with long gaps (rather than long sequences). Therefore, we selected baselines tailored to handle irregular models. Traditional time series models struggle with the sparsity and irregularity of the data. For instance, our baseline TSMixer [1] demonstrates superior abilities to leverage longer sequences and better generalization than other multivariate models but didn't perform well on irregular samples. However, we acknowledge that the third reference you suggested (Primenet [2])  addresses similar issues. We agree that it could provide further insights into the performance of our proposed model. However, we would like to address the constraints we are facing with the timeline. The model you suggested was not available during the experimentation phase of our proposed model, and we have limited time to incorporate it into our study. However, we are aiming to add it, hoping it will be ready soon. Additionally, this work will be also addressed in the Related Work section.
>
> [1] Chen, S.A., Li, C.L., Yoder, N., Arik, S.O. and Pfister, T., 2023. Tsmixer: An all-mlp architecture for time series forecasting. arXiv preprint arXiv:2303.06053.
>
> [2] Chowdhury, Ranak Roy, et al. "Primenet: Pre-training for irregular multivariate time series." Proceedings of the AAAI Conference on Artificial Intelligence. Vol. 37. No. 6. 2023.

---

> ### Author Response · Authors · 2024-05-07
> **Key Revision Highlights**
>
> Dear Reviewer,
>
> We appreciate your invaluable feedback, and we've incorporated your suggestions into our manuscript. Below are the key revisions made based on your suggestions:
>
> 1. **More related work**: We have expanded the related work section to encompass the latest research. Additionally, we highlight the distinctions between our contributions and the focus of other recent models. This update can be found on Page 3.
>
>
> 2. **Notation and Symbols**: To enhance clarity, we have revised the notation and symbols, spanning Pages, 5,6, 7, 8 and 9 based on your suggestions, including a clear description of Equations 4 and 5, $u$, $t_i$ and $h$, as well as revising Equations 8 and  9, and Algorithm 1.
>
> 3. **Figure 2**: Figure 2 has been changed, with explicit labelling of inputs and outputs for each module, detailed on Page 6.
>
> 4. **Model Overview**: We've added subsection 4.2, "Model Overview," providing a comprehensive description of the model and the input-output of each part, located on Page 6.
>
>
>
> Thank you once again for your insightful recommendations.

---

> ### Comment · Reviewer_kkXW · 2024-05-16
> **A few more clarification requests on notations.**
>
> I would like to thank the author for making the changes and improving the clarity of the presentation. The purpose and functionality of each component in the proposed approach is much clearer to me and I will take the changes into my consideration in the review. However, there are still some confusing details on the notation and they are worthy of further clarification.
>
> 1. At the end of Sec. 4.3, the author used the notation $U=u_1, u_2,\dots,u_k$ to represent a set of hidden states for each sequence. However, this is a non-standard notation for a set. There are also two different ways of interpreting $u_i$ here. It could mean the hidden states for the $i$th sequence or the hidden state at $t_i$ according to Eq. 4 and 5. The same notations is also used in the caption of Figure 2.
> 2. At the end of Page 7, $\\{l_1=\\{u_i\subset U\\},\dots, l_L=\\{u_i\subset U\\}\\}$ is used to represent a set of latent representations. It seems that $u_i$ is overloaded with a different definition in this notation. Is this set the output of the pyramidal sorting?

---

> ### Author Response · Authors · 2024-05-17
>
> Thank you for your continuous feedback and comments. Your feedback has been invaluable in improving our work. We apologize for any misunderstanding regarding the notation used to represent the set of hidden states.
>
> To address this, we have revised the notation to accurately reflect the set of hidden states for each sequence. The corrected notation is: [ $U = \{\{u_{i}^{(1)}\}_{i=1}^{n}, \{u_{i}^{(2)}\}_{i=1}^{n}, \cdots, \{u_{i}^{(k)}\}_{i=1}^{n} \}$ ], Please see Page 7 for the finale Formula.
>
> This notation appropriately represents the set of hidden states for each sequence $(k)$ at all time points. The hidden state at a particular time point $\(t_i\)$ is represented as Equations 4 and 5. Additionally, we have revised Figure 2.
>
> Regarding your second point, each $(l)$ is a subset of the learned representations set $U$. So $(l)$ includes some of the $(U)$ based on their relation for the samples. This was adjusted as follows (on the last paragraph of  page 7) :
>
> "The assigned importance weights $( \alpha_x )$ are used to generate a set of hierarchical levels of related latent representations as {$ l_1, l_2, \dots, l_L $}, where each $( l_j)$ represents a subset of the hidden states from $(U)$, i.e., $ l_j =${ $u \subseteq U$ }. "
>
> Once again, we appreciate your feedback, which has contributed significantly to the refinement of our work.

---

### Review · Reviewer_1dTu · 2024-04-30

**Summary Of Contributions:**

The authors introduce a neural network designed to enhance the robustness of sequence representations, which operates regardless of sequence length or data sparsity level. It uses other training sequences to enrich the latent representation that corresponds to an input sequence using an attention mechanism between the input sequence and other sequences. The model is compared against standard neural ODE variants on standard benchmarks, and performs comparatively well.

**Audience:**

Yes

**Claims And Evidence:**

Yes

**Requested Changes:**

- The works listed in the first weakness above should be discussed.
- Caption of Figure 1c should be elaborated. Is the data "bumpy" or "intermittent"?
- Symbols and notations table should be revisited, e.g., $K$ does not denote a set but set size, using $\mathbf{x}$ to denote a vector of observations instead of $x$, why $Link − ODE$ instead of a single letter, etc.
- What exactly is the cut_out function? What are its hyperparameters?
- $f\theta$ in (4) and $arg_\rho min$ are incorrect notation.
- How is $u$ obtained? Does it correspond to $h$ in the encoder?
- What does $(t_{i-1},...t_i)$ mean in (4)?

**Strengths And Weaknesses:**

### Strengths
- The approach relies on using multiple sequences for modeling an unseen trajectory, which to my knowledge is a novel aspect.
- Strong results.

### Weaknesses
- One of the main claims of the paper (*"ODEs describe the evolution in time of a process that depends on one variable (initial condition) (Chen et al., 2018); hence, for ODE-based models, the continuous trajectories of the hidden state are described by just one variable"*) should be revisited. Although this holds for vanilla NODEs (Chen et al., 2018), upcoming works [1,2] partially addressed this. So, I'm not sure if *"ODEs are effective but may not provide the optimal representation for the whole sequence"*.
- I don't clearly see how the problem of "ODE models relying on a single initial state" can be alleviated by "more generalised hidden trajectories". More specifically, what exactly does "using multiple sequences" bring?
- Writing and presentation should be improved (see requested changes).
- As far as I can see, hyperparameter selection is not discussed and some ablations are missing (e.g., how does the performance change with a non-linear decoder inside the encoder or with the number of training sequences considered in pyramidal attention, computational complexity and execution time of the proposed approach

[1] Neural controlled differential equations for irregular time series

[2] Latent Neural ODEs with Sparse Bayesian Multiple Shooting

---

> ### Author Response · Authors · 2024-05-05
>
> Thank you for your feedback. Below are responses addressing all your concerns and questions
>
> - Weakness 1:
>
> In Neural Controlled Differential Equations (CDE) [1], the authors address the issue of how to modify the hidden trajectories based on newly received data, which is slightly different from the issue we are addressing. Our work focuses on the challenge of model stability when there are significant time gaps between observations and whether the  ODE representation is still effective. Even for ODE-RNN, the model updates the hidden state for the newly received data in a different way than CDE.  In CDE and ODE-RNN, that representation depends on the last observed values until there is a new observation. However, both will follow the last observation until the next is available. We will make sure to make this clear in the text.
>
>
> Similarly, for this second work [2], their focus is not on having a long gap; rather, it is on having a long sequence. However, this is a very recent work that was not published by the time we finalised our work. We will make sure to discuss their model in our paper. In summary, the proposed model works by dividing the time grid into blocks and having multiple ODE representations on various shots to have faster training
>
> - Weakness 2:
>
> The primary issue we're addressing is the stability of ODE-based methods when there are long time gaps between observations. This challenge arises from the fact that the model relies on the last state over long intervals, whether it's the initial state or the last observed state (which subsequently becomes the initial state for the current ODE solver).
>
> This hidden state becomes constrained as the ODE predicts the dynamic behaviour of trajectories without any actual observations for extended periods due to the large gaps between observations. Consequently, we aim to leverage information collected from other related samples at a particular time point to improve the model's ability to infer more accurate hidden trajectories for unavailable observations, even when data gaps are long.
>
> - Weakness 3:
>
> We have addressed these issues. Thank you for pointing them out.
>
> - Weakness 4:
>
> The model hyperparameter was discussed in section 5.2 (Setup); we selected the hyperparameter that yielded the best performance for the original ODE-RNN. For the auto-encoder, we used ODE-RNN for the encoder and a shallow sequential network of one linear layer to decode the data to the data space and solve back the ODE. For Link-ODE, we used the ODE function for one hidden layer and 100 units. The latent dimension used was 10 for all data sets. The fifth-order (dopri5) solver is used for the ODE, and 200 as batch size."
>
>
> The number of training sequences is different across different data sets; we will make sure to include the dataset statistics; however, as we are not excluding any of the samples, the pyramidal attention includes all the samples. On the other hand, the more related the sample, the higher the weight will be assigned to it. This is important to avoid bias on individual samples and reduce the influence of outliers or noise.
>
> Finally, the computational complexity is similar to the ODE-RNN. However, there is an additional cost of using the pyramid attention, which is the same as the standard operations of the attention mechanism.
>
>
>
>
> - Requested Changes:
>
> A- This recent work will be addressed in our paper as mentioned in the response for Weaknesses 1.
>
>
> B- The result in Figure 1.C is an intermittent irregular time series with a diverse level of sparsity; we will make sure that this is clear in the caption.
>
>
> C- All symbols and notations will be revised and fixed.
>
> D- The cut-out function is just a simple function that randomly removes samples from the dataset to generate irregular data. This function has been deployed in several related works. Particularly, it takes time series data and removes some of the values based on a specific ratio. In our case, 30\% of the real world dataset was removed, while 10\% to 40\% of the synthetic data were removed.  We will make sure this function is clear in the article.
>
>
> E-  yeah, $u$ is the hidden state; we represent it as $u$ to distinguish between the hidden state from the final ODE-Link and the ODE autoencoder. We will make sure to make it clear and consistent.
>
> F- $t$ is a time point in the sequence, $t_{i-1}$ is the last time point of available observation, and $t_i$ is the current time point; the continuous hidden state between these time points is calculated based on ODE as there is no observation.

---

> ### Author Response · Authors · 2024-05-07
> **Key Revision Highlights**
>
> Dear Reviewer,
>
> We appreciate your invaluable feedback, and we've incorporated your suggestions into our manuscript. Below are the key revisions made:
>
> 1. **More related work**: We have expanded the related work section to encompass the latest research. Additionally, we highlight the distinctions between our contributions and the focus of other recent models. This update can be found on Page 3.
>
>
> 1. **Figure 1.C**: The caption for Figure 1 has been updated for clarity, along with a description of the dataset within the main text; this can be found in Section 1, paragraph 3, on Page 2.
>
> 2. **Notation and Symbols**: To enhance clarity, we have revised the notation and symbols, spanning Pages, 5,6, 7, 8 and 9 based on your suggestions, including a clear description of Equations 4 and 5, $u$, $t_i$ and $h$.
>
> 3. **Cut-out Function**: The cut-out function has been described in Section 4.3,  Page 7.
>
>
> 4. **Model Overview**: We've added subsection 4.2, "Model Overview," providing a comprehensive description of the model and the input-output of each part, located on Page 6.
>
>
>
> Thank you once again for your insightful recommendations.

---

### Review · Reviewer_AfiC · 2024-05-02

**Summary Of Contributions:**

The paper proposes a new neural ODE-based architecture for forecasting partially observed time series. The architecture extends the popular ODE-RNN model in several ways. First, an unsupervised autoencoder model is trained to reconstruct the missing time series values. Then, the representations for each time step learned by the autoencoder are passed through a pyramidal attention block to exchange information across the time series. The proposed architecture, called Link-ODE, is compared with various popular models for irregular time series forecasting on 4 real-world and 3 synthetic datasets. The proposed Link-ODE model provides consistent improvements over the base ODE-RNN model in terms of prediction quality, as measured by the MSE and ROC AUC metrics.

**Audience:**

Yes

**Claims And Evidence:**

No

**Requested Changes:**

1. (critical) Please provide a clearer explanation and motivation for the learned embeddings and the pyramidal attention in the model architecture. Here are some questions that I would like to see answered:
	- Are the embeddings $u$ free model parameters, or can they be computed inductively for new time series not seen at training time?
	- What is the purpose of the embeddings and the pyramidal attention? Is it to capture the interactions between distinct time series / samples (e.g., time step $i$ of sample $k$ may influence the time step $i^\prime$ of sample $k^\prime$?)? Why is capturing such interactions helpful for prediction if the time series are unrelated (e.g., time series corresponding to different patients in a medical dataset)?
2. (critical) Please update central + right parts of Figure 2 as well as the corresponding equations, indicating precisely what are the input and output for each model component. Some inconsistencies and unanswered questions in the current figure that greatly hurt readability:
	- According to Equation 5, denoising autoencoder produces the hidden states $h_i$ as output, but the figure shows that the autoencoder returns representations $u$ as output
	- Indices in $u_{xi}, ..., u_{xn}$ probably contain typos
	- How are $l_1, ..., l_{|L|}$, $U$ defined?
	- What exactly is the input & output to pyramid attention?
	- Use consistent notation for indexing across individual items, time steps, and dimensions
	- How is the output of the Link-ODE layer used to compute the final objective function and train the model?
3. (critical) Please adjust the claims related to the "restricted hidden trajectories" in ODE-based models, or provide a more thorough analysis supporting this claim.
4. (recommended) The current ablations do not test all the novel architectural pieces introduced in this paper. For example, and even more informative ablation would be to directly use the hidden states $h_i$ produced by the autoencoder to predict the time series values, without using the sequence embeddings $u$ at all.

**Strengths And Weaknesses:**

### Strengths
**Relevant problem**: The paper studies a highly relevant problem of forecasting intermittent and partially observed time series. These are common in real-world applications, but are not often considered in ML literature.

**Strong empirical results:** The architecture proposed in the paper provides a consistent improvement over the ODE-RNN model, as well as other popular time series models, in terms of the prediction accuracy.
### Weaknesses:
**1. Presentation clarity**: The paper is missing the motivation for some critical model components (learned representations $u$ and the pyramidal attention block), and their precise definition is unclear. For example, according to Section 4.2, the ODE autoencoder produces for each time series $x \in \mathbb{R}^n \times D$ a sequence of hidden states $\{h_1, ..., h_n\}$. However, according to Figure 2, the output of the autoencoder are the representations $u_{xi}, ..., u_{xn}$ for the individual time series, which contradicts Section 4.2. The meaning of the indexing in $u_{xi}, ..., u_{xn}$ is also unclear here.

Moreover, the meaning of certain symbols ($f$, $\theta$, $x_i$, $h_i$, $w$, $L$) is overloaded multiple times throughout the paper, which makes it hard to follow. For instance, $h_i$ is used to denote both the output of the ODE autoencoder as well as the output of the Link-ODE layer. Other symbols such as $l_i$ seem to be undefined, and are only used as input in Equation 10.

**2. Unsupported claims**: The first of the claimed main contributions ("We have demonstrated how ODE-based models may provide inaccurate and restricted hidden trajectories, as they are affected by the length of the time lapse between observations") is not fully supported by the results presented in the paper. In fact, the results shows in Table 4 only demonstrate that the performance of the ODE-RNN model degrades as the fraction of missing data increases. However, this could be explained by other factors - such as the decreasing amount of training data. The latter hypothesis would be in line with other results shown in Table 4, as non-ODE-based models like RNN also show degrading performance as the fraction of missing data increases. I believe that a more careful investigation is necessary to support the claim about the restricted hidden trajectories.

More generally, the claims about the behavior of different models as sparsity is changing are based exclusively on synthetic gaussian datasets with periodic time series. It should be rather straightforward to extend this analysis to real-world data by introducing sparsity/missing values, which would strengthen the claim.

### Minor details
- In Figure 1, it's unclear where missing values correspond to observed zeros or unobserved values. A better idea would be to use a line plot with marks at observed/unobserved values. Also, the Y-axis is unlabeled.
- There are numerous small typos in typesetting that hurt readability, such as $Xs$ instead of $X_s$, $arg_\rho min$ instead of $\arg \min \rho$, $l.w$ instead of $l \cdot w$ - please consider proofreading the paper and fixing them.

---

> ### Author Response · Authors · 2024-05-05
> **Discussing the Weaknesses**
>
> Thank you for your feedback. Below are responses addressing all your concerns and questions. We will start by discussing the weaknesses you addressed, which will possibly answer your questions. Following that, we have addressed the minor details and, finally, the requested changes.
>
> - Weakness 1: Presentation clarity
>
> The learned representations are hidden representations for samples in the data. It is used to generalise the final hidden state for the irregular sample. Imagine a patient record $(patient\_1)$; usually, the hidden state of a one time point of a sample is defined using its last hidden state and the value at the current time point, as sometimes this value is not available for long periods of time due to many reasons (i.e. the patient record is irregular). An ODE solver is used to provide a continuous representation regardless of whether the data is available (as described in section 3 ). However, ODE also depends on the last available representation seen for that patient, $(patient\_1)$;  this will generate a restricted and not reliable representation for the gap, especially if the sequence is very long and very sparse (as seen in the results for the motivation in Figure 1). In SeqLink, the model learns the representation for other samples (other patients) and allows the ODE solver to give a continuous representation for unavailable time points in $(patient\_1)$ by considering a more general representation from other samples in the data. So, the learned representation is a source of information that could be useful to provide a better representation of sparse and irregular samples.
>
> Following that, as not all patients will be helpful and related to the current patient $(patient\_1)$, the pyramid block allows us to focus on the information that would be most helpful and more related by giving it a higher weight.
>
> Regarding the notations, we apologise for the unclear details. We refer to the hidden state (the learned representation) of a sample as $h_1,\cdots,h_n$, which is the most common notation used to represent the hidden state in deep learning. However, since we are considering several representations, we are referring to the one generated by the ODE auto-encoder as $u_{1}, u_{2},\cdots, u_{n}$ for each time series $k$.  We have fixed that in Equations 4 and 5. We also improved the text to be clear and more readable. In summary, $U = {u_{1}, u_{2},\cdots, u_{k}}$ is the learn representation for the ODE auto-encoder for sample $k$. Meanwhile, $h$ is used to represent the last hidden state of the model, which is the output of the Link-ODE layer. Thanks for pointing this out.
>
> Regarding  $l$,  we make sure it is defined in section 4.4. In summary, $L$ is the hierarchical level created by the pyramid module as $L ={l_1, l_2, \cdots, l_L}$ based on the number of levels $(L)$. Each element $l$  has a subset of the learned representations as $l=$ { $u_k \mid k \in K $ }.
>
>
> -  Weakness 2: Unsupported claims
>
> In this statement, "We have demonstrated how ODE-based models may provide inaccurate and restricted hidden trajectories, as they are affected by the length of the time lapse between observations", we are referring mainly to the performance of the original ODE-based models, which has been shown as motivation in Figure 1 and was supported by the experiments in section 5.3 and Table 6 as well as the case study in Figure 5. This is fixed to make it clear. In summary, we change the claim to: "We have demonstrated how traditional ODE-based models may yield inaccurate predictions and less effective hidden representation when applied to data with a high level of sparsity. Additionally, we show how they are affected by the length of the time lapse between observations. These experiments also serve as the motivation for this work."  We will also emphasise that in the experiments.
>
> - Minor details:
>
> 1 - Figure 1: Thanks for your suggestion; this Figure has been fixed according to your comments.
>
> 2- Typos: The typos of the paper have been fixed.
>
> In the next comment, we will address the Requested Changes.

---

> ### Author Response · Authors · 2024-05-05
> **Requested Changes**
>
> - Requested Changes 1:
>
> We improve the explanation and motivation for the learned embeddings and the pyramidal attention along with the notation as the response of (Weakness 1).
>
> A- In the case of new time series data, it needs to be mapped to the most related representation of the other samples. Therefore, the embeddings of this sequence for the new time series will be computed.
>
> B- Yeah, that's true. It will help produce a final hidden representation of the gap in the current sample. In case the data is not related, it will automatically get a very low weight by the pyramid module, so it will not influence the final representation.
>
> - Requested Changes 2:
>
> The Figure and the Equations are improved to represent the correct input and output as discussed in the response of (Weakness 1).
>
> A-  $h_1,\cdots,h_n$ will be used to represent the final hidden representation.
>
> B- $U$ is the learn representation for the ODE auto-encoder.
>
> C- $L$ is the hierarchical level created by the pyramid module as $L ={l_1, l_2, \cdots, l_L}$ based on the number of levels $(L)$. Each element $l$ in $L$ has a subset of the learned representations $U$ as $l= $ { $u_k \mid k \in K $ }. $U$ is a vector that represents all learned states.
>
> D- The input and output will be clarified in both the Equations and Figure 2. In summary, the input of the pyramid attention consists of the learned representations from the ODE auto-encoder, while the output is a sorted embedding based on the relations between the samples.
>
> E- All notations and symbols are reviewed and fixed.
>
> F- The output of the Link-ODE layer is a representation of the sparse data at each time point. This representation contains the learned relationships and dynamics within the data. To compute the final objective function and train the model, this representation is fed into a standard output layer that is defined based on the task at hand, whether it be forecasting or classification. The output layer in a forecasting scenario is a dense layer of linear activation function to predict future values. Alternatively, in a classification task, the output layer is a dense layer with a softmax activation function to assign samples to different class labels.

---

> ### Author Response · Authors · 2024-05-07
> **Key Revision Highlights**
>
> Dear Reviewer,
>
> We appreciate your invaluable feedback, and we've incorporated your suggestions into our manuscript. Below are the key revisions made:
>
>
> 1. **Figure 1**: We've updated Figure 1 to show the unobserved data, now highlighted with yellow hatching on Page 2.
>
> 2. **Notation and Symbols**: To enhance clarity, we have revised the notation and symbols on Pages 7 and 8.
>
> 3. **Equations in Section 4.3**: All equations in Section 4.3 have been revised, now presented on Page 7.
>
> 4. **Figure 2**: Figure 2 has been changed, with explicit labelling of inputs and outputs for each module, as shown on Page 6.
>
> 5. **Model Overview**: We've added subsection 4.2, "Model Overview," providing a comprehensive description of the model and the input-output of each part, located on Page 6.
>
> 6. **Claim 1 Clarification**: Claim 1 in the introduction (contributions - Page 2) has been clarified and emphasized, both in the experiment (Section 5.3, paragraph 4, Page 12) and in the case study (Section 6, Page 15).
>
>
> 7. **Final Layer**: The final layer of the model, to generate the prediction, is now described in the last part of Section 4.5, found on Page 9.
>
> 8. **Ablation Study Clarification**: We apologize for any confusion regarding the ablation study. Your suggestion is, in fact, the first configuration of the ablation study. Specifically, in the "Unified hidden trajectories" configuration, we rely solely on the hidden state from the autoencoder without incorporating embedding, as the importance rates were not utilized. This clarification is provided in subsection 5.4.
>
>
> Thank you once again for your insightful recommendations.

---

### Decision · Action_Editor_zbWz · 2024-06-17

**Recommendation:** Accept with minor revision

**Comment:**

Please carefully read through your submission and fix typos and language errors, there are some left as pointed out by Reviewer AfiC.
Just a few examples "A neural ODEs"; "Brouwer et al." (cited authors's name is De Brouwer as it is visible in the citation few words later),
"ODEs models", etc.

Please change "Our code will be available soon" in the camera ready version with a reference to the Git repo.

**Audience:**

Yes

**Claims And Evidence:**

Yes, the claims are supported by evidence.